# Fabrication of EP@PDMS@F-SiO_2_ Superhydrophobic Composite Coating on Titanium Alloy Substrate

**DOI:** 10.3390/biomimetics10060404

**Published:** 2025-06-16

**Authors:** Chaoming Huang, Jinhe Qi, Jie Li, Xinchi Li, Jiawei Chen, Shuo Fu, Yanning Lu

**Affiliations:** 1College of Marine Engineering, Jimei University, Xiamen 361021, China; hchaoming@jmu.edu.cn; 2Fujian Institute of Innovation for Marine Equipment Detection and Remanufacturing Industrial Technology, Xiamen 361021, China; 3School of Computer and Artificial Intelligence, Beijing Technology and Business University, Beijing 100048, China

**Keywords:** titanium alloy, laser etching, EP@PDMS@F-SiO_2_ composite coating, clean up, UV durability

## Abstract

In this study, a preparation method of superhydrophobic composite coating based on a titanium alloy (Ti-6Al-4V) substrate is proposed. The micro-scale pit array structure was fabricated via laser etching technology. Utilizing the synergistic effects of epoxy resin (EP), polydimethylsiloxane (PDMS), and fluorinated nanosilica (F-SiO_2_), we successfully prepared an EP@PDMS@F-SiO_2_ composite coating. The effects of the contents of EP, PDMS, and F-SiO_2_ on the surface wettability, mechanical stability, and UV durability were studied by optimizing the coating ratio through orthogonal experiments. The results show that the micro–nano composite structure formed by laser etching can effectively fix the coating particles and provide excellent superhydrophobicity on the surface. The coating retains high hydrophobicity after paper abrasion (1000 cm under a 200 g load), demonstrating the mechanical stability of the armor-like structure, High-content F-SiO_2_ coatings exhibit greater UV durability. In addition, the coating surface has low droplet adhesion and self-cleaning capabilities for efficient contaminant removal. The research provides theoretical and technical support for the design and engineering application of a non-fluorinated, environmentally friendly superhydrophobic coating.

## 1. Introduction

With the development of modern industry and high technology, the demand for the surface functionality of materials is increasing. Superhydrophobic surfaces have excellent ice resistance [1,2], self-cleaning [3,4], corrosion resistance [5], and drag reduction characteristics [6,7], which have attracted extensive attention in aerospace [8], marine engineering [9], biomedicine [10], and other fields in recent years. Because the surface typically relies on the interaction between a complex micro/nano multi-scale structure and low-surface-energy substances, once it is affected or damaged by the external environment (such as abrasion, UV radiation, corrosive medium, etc.), it is prone to sudden performance degradation, failure, and other problems. Therefore, it is of great significance to study surface modification strategies with superhydrophobicity and good durability to improve the performance of metal surfaces.

Titanium alloy, as a metal structural material with excellent characteristics such as light weight, high strength, and corrosion resistance, has played an essential role in aviation, aerospace, and marine equipment [11,12,13]. A durable and stable superhydrophobic coating on its surface is expected to enhance further the performance of anti-icing, drag reduction, and anti-abrasion in harsh service environments. Current standard methods for fabricating superhydrophobic surfaces include laser etching [14], chemical etching [15], electrochemical anodic oxidation [16], sol-gel processing [17], and electrospinning [18]. For instance, Vercillo et al. used laser processing on Ti6Al4V, achieving a WCA of 163°, 50% ice adhesion reduction, and durability over 16 icing cycles [14]. Wu et al. combined anodizing and H_2_O_2_ treatment to obtain a WCA of 151.9°, with good biocompatibility, but the operation is complex [16]. Czyzyk et al. applied sol-gel with double curing to reach a WCA of 160° and nearly 0° SA, enduring tape peeling and 1000 h UV aging [17]. Among them, laser etching has the advantages of good controllability, high processing efficiency, and wide application range. It can accurately prepare multi-scale micro/nanostructures on the surface of titanium alloy, and combine with the subsequent nanoparticle composite coating technology, to improve significantly the wettability control effect of the material surface. Furthermore, laser processing technology has attracted increasing attention in the field of biomimetic manufacturing due to its cross-scale fabrication capabilities, broad material compatibility, operational simplicity, and high efficiency [19,20]. Laser subtractive manufacturing based on ablation and induction effects offers a highly precise and versatile approach for fabricating large-area biomimetic micro/nanostructured surfaces [21].

There is a wide variety of nanomaterials, but in the field of superhydrophobic applications, silica (SiO_2_) nanoparticles and their modified materials demonstrate significant advantages over carbon-based nanomaterials [22] and polymer-based nanomaterials [23]. After modification, their visible light transmittance can reach 85–92% [24], making them particularly suitable for optical applications like solar panels. Their unique micro/nanostructure provides excellent corrosion resistance, abrasion resistance, and mechanical stability, which significantly outperforms carbon-based materials. After modification with fluorosilane, the contact angle of the material can reach 150–170°, while also possessing both self-cleaning and efficient oil–water separation functions [25]. Additionally, the raw materials are widely available, and the modification process does not require fluorine, making the preparation process simple and easy to implement.

Due to its high hardness, good abrasion resistance, and stable chemical properties, nano-SiO_2_ particles are often selected as the skeleton material or functional filler of superhydrophobic coating. Ke et al. prepared a layered film based on PDMS and SiO_2_ by a simple drop-coating method [26]. The modified SiO_2_ not only improved the dispersion of particles, but also improved the compatibility with PDMS. PDMS has excellent characteristics such as low surface energy and thermal stability, which is favored by researchers in hydrophobic material preparation. He et al. reported a simple one-step 3D printing method, printing PDMS into an ordered porous structure to produce a special wetted film surface [27]. Anisotropic humidity was achieved by adjusting the structure of submillimeter holes, and the test showed good thermal aging resistance. However, it is often tricky for SiO_2_ particles and PDMS to bond firmly to the metal substrate, so EP was introduced to improve the adhesion of the nanoparticle coating. For example, Li et al. studied the preparation of a honeycomb epoxy resin structure by the PDMS template method. Then, they sprayed hydrophobic SiO_2_ nanoparticles to prepare a high transparency and superhydrophobic honeycomb structure coating [28]. Although hydrophobic surface modification can also be achieved by using EP or PDMS alone to a certain extent, they can often produce significant gain effects when combined with SiO_2_ nanoparticles and coated on the titanium alloy substrate treated by laser texture [29].

Based on the above ideas, this study focuses on laser etching of microstructure-SiO_2_ nanoparticles-EP@PDMS. The superhydrophobic surface of the titanium alloy with micro–nano composite structure was successfully prepared by the composite method. This non-fluorinated material shows significant characteristics such as environmental friendliness and economic advantages in the preparation process. The test results show that the prepared titanium alloy composite coating surface has good hydrophobicity, and the armor structure etched by laser has good mechanical stability, which can also maintain the hydrophobicity of the surface after sandpaper abrasion. In addition, the adhesion of the surface to droplets is very low, which provides excellent self-cleaning ability and a decontamination effect. This study not only confirmed the synergistic effect of EP and PDMS in nanoparticles, but also provided an essential reference for the further promotion of durable superhydrophobic coatings in large-scale industrial production.

## 2. Experiment

### 2.1. Materials

The substrate used in the experiment was titanium alloy (Ti-6Al-4V) with a size of 20 × 20 × 3 mm^3^, which was purchased from Dongguan Guanyue metal materials Co., Ltd. (Dongguan, China). E44 EP and amine curing agent (aliphatic, amine value: 600~700 mgKOH/g) were purchased from Shanghai Maclean Biochemical Technology Co., Ltd. (Shanghai, China) PDMS (sylgard 184) and curing agent were purchased from Dow Chemical Company (Midland, MI, USA). Nano SiO_2_ (particle size 20–30 nm), ethyl acetate (EA, 99.5%), anhydrous ethanol (99.7%), 1H, 1H, 2H, 2H-perfluorodecyltriethoxysilane (molecular formula C_16_F_17_H_19_O_3_Si, FAS, 97%), and γ-aminopropyl triethoxysilane (silane coupling agent KH550, 99%) were purchased from Beijing InnoChem Science & Technology Co., Ltd. (Beijing, China) Ultrapure water was prepared in the laboratory. All chemical reagents were analytical grade reagents.

### 2.2. Preparation of Superhydrophobic Coating

#### 2.2.1. Preparation of F-SiO_2_

As shown in Appendix A, the superhydrophobicity of silica is clearly demonstrated by the distinct contact angles formed by water droplets on the silica particle-coated surface, indicating excellent water-repellent performance. To improve the dispersion of SiO_2_ in ethyl acetate (EA) solution and enhance its interfacial adhesion and mechanical compatibility with PDMS and EP, surface modification of SiO_2_ using FAS is required. The modification procedure consists of three main steps: dispersion and soaking, drying, and grinding. Specifically, SiO_2_ particles were added to a 1 wt% FAS ethanol solution, followed by stirring and ultrasonic dispersion. The solution was then left to stand at room temperature for 1 h. Afterward, it was dried in an oven to fully evaporate the solvent, and the resulting solid was ground using a mortar to ensure a uniform dispersion of the SiO_2_ particles, yielding fluorinated silica (F-SiO_2_).

As shown in Appendix A, the enhanced hydrophobicity of the F-SiO_2_ is attributed to the hierarchical micro/nanostructured roughness and the low surface energy introduced by fluorination, which together promote the formation of a stable Cassie–Baxter state. This state minimizes the solid–liquid contact area, reduces surface adhesion, and imparts excellent self-cleaning properties.

#### 2.2.2. EP@PDMS @Preparation of F-SiO_2_ Coating Solution

In this experiment, the orthogonal design table (three factors and six levels) was used to explore the optimal ratio for improving the surface hydrophobicity of titanium alloy by controlling the amounts of EP, PDMS, and SiO_2_. The mixed solution in this experiment is divided into A and B. Solution A is the bottom EP layer solution, and solution B is the upper EP@PDMS@F-SiO_2_ layer mixed solution. Next, the specific preparation methods of two different solutions are introduced. The preparation of solution A is as follows: add 1.4 g EP and 0.35 g amine curing agent to 10 g EA for stirring. The preparation of solution B is as follows: EP, PDMS, F-SiO_2_, and the corresponding proportion of curing agent were added to EA (EP: amine curing agent = 4:1, PDMS: curing agent = 10:1, both mass ratios), 0.1 g KH550 was added, and then stirring and ultrasound were carried out successively to obtain a uniform mixed solution. The specific proportion of solution B is shown in Table 1.

#### 2.2.3. Preparation of Superhydrophobic Titanium Alloy Specimens

Firstly, the sample is polished with #400, 800, 1200, and 1500 grit sandpaper to ensure that the surface is as smooth as possible. Next, the square groove array microstructure is generated by controlling the laser beam ablation on the surface, which is also cleaned after laser etching to remove surface impurities. Then, after laser etching, the sample is immersed in solution A for 10 min, and then taken out and put into the oven for preliminary drying. Then, the sample is placed into solution B, soaked for 10 min, and then taken out to deposit SiO_2_ particles into the square groove of laser etching. Finally, the soaked sample is taken out and put into the oven to dry at 120 °C for 2 h. Figure 1 is a schematic diagram of the preparation of the titanium alloy superhydrophobic coating surface. Figure 1a is the shape, and structural parameters of the laser etching sample surface, Figure 1b is the preparation process of solutions A and B, and Figure 1c is the schematic diagram of the coating attached to the titanium alloy micro textured surface.

### 2.3. Characterization of Surface Properties

A scanning electron microscope (phenom XL, phenom scientific, SEM, Thermo Fisher Scientific, Bleiswijk, The Netherlands) was used to observe the surface morphology of the sample, and an energy dispersive spectrometer (EDS) was used to analyze the surface element composition of the sample. A laser confocal microscope (SZ-2000, ASI, Eugene, OR, USA) was used to construct the three-dimensional profile of the sample surface etched by the laser and analyze the depth of square pits. The WCA was measured using an optical contact angle goniometer (XG-CAM, XYCXIE, Shanghai, China). Measurements were performed three times using 2 μL of ultrapure water at different locations on the sample surface, and the average value was reported. Similarly, the SA was measured three times at different positions on the same surface using 2 μL droplets, and the average value was taken as the final result. A UV lamp (150 W) was used to irradiate the sample to simulate the aging effect of UV on the surface coating in sunlight.

## 3. Surface Property Test and Characterization

### 3.1. Surface Morphology and Element Composition

Laser etching technology is a non-contact processing technology with a small heat-affected zone and high precision. To obtain a solid and durable hydrophobic surface of titanium alloy, the surface of the titanium alloy was laser etched to form square array pits, and the coating particles were embedded in the pits to create a multi-scale micro–nanocomposite surface. Before SEM observation, the sample surface was coated with a thin layer of gold to form a conductive layer, thereby enhancing electron conductivity and image clarity. SEM imaging was conducted under high vacuum conditions with an acceleration voltage of 15 kV.

As shown in Figure 2(a1,a2), SEM observations reveal that the surface of the titanium alloy after laser etching is characterized by the presence of five small circular pits. The circular pits are formed by the rapid melting and vaporization of the material by the high energy density of the laser beam. The laser scanning speed is relatively fast without overlapping the adjacent laser beams, resulting in such discontinuous circular pits. However, there are obvious sputtered micron-sized particle bulges around the square pits, as shown in Figure 2(a3). This is due to the rapid cooling and solidification of the splashed liquid metal during the melting and vaporization process when the laser beam acts on the surface of the titanium alloy. Therefore, the micro-scale surface structure was formed on the surface of the titanium alloy by laser etching.

After observing the surface after coating treatment, as shown in Figure 2(b1–b3,c1–c3), nanoparticles cover the sample’s surface, and the pits formed by laser etching effectively fix and contain the coating particles. According to the concentration of SiO_2_ particles in the coating solution, the degree of deposition on the surface after etching also differs. As shown in Figure 2(b2), the surface of Sample 12 is covered by SiO_2_ particles. These particles are not only filled in the square pits, but also exist on the surface that is not etched between adjacent pits, as shown in Figure 2(b1). There are particles with irregular size and shape, which may be formed due to the clustering of SiO_2_ particles during fluorination treatment and incomplete dispersion during grinding after drying. As shown in Figure 2(b3), it can be seen that SiO_2_ particles are bonded to the substrate surface, while PDMS and EP in the coating bond SiO_2_ particles to form a nanoscale structure. The surface of Sample 1 is shown in Figure 2(c1,c2). Due to the low concentration of SiO_2_ particles, the area covered by the surface is relatively small, and only part of the pits and surfaces are embedded. In Figure 2(c3), SiO_2_ particles are wrapped and adhered to the substrate by PDMS and EP. Appendix A shows a three-dimensional representation of Sample 3, illustrating the surface morphology of the titanium alloy after the application of the superhydrophobic coating. As observed, the silica (SiO_2_) particles are uniformly distributed across the surface, forming a complex hierarchical micro/nanostructure. It can be shown that the microstructure formed by laser etching and the filling of nanoparticles work together to construct the surface of this micro–nano composite structure.

The square groove array was selected in this study due to its geometric symmetry, which facilitates a uniform deposition of the coating and enhances mechanical compatibility with the composite structure. As shown in Figure 3a, the laser-etched surface of the titanium alloy sample displays a distinct and regular groove structure. In Figure 3b, the cross-sectional profile corresponding to the line segment in Figure 3a is obtained via laser confocal microscopy. The measured groove depth is approximately 150 μm, which is adequate for an effective embedding of SiO_2_ nanoparticles and contributes to the formation of a stable composite coating structure. As illustrated in Figure 3c, the coated sample’s surface morphology clearly indicates the successful integration of SiO_2_ particles into the grooves, leading to a marked increase in surface roughness and the attainment of outstanding superhydrophobic performance.

EDS analysis was performed on the surface of the coated specimen, as shown in Figure 3a and Figure 4b. The surface of the coating is mainly composed of O, Si, Ti, and C elements. In Figure 4a, the most distributed element is O, accounting for 53.2%, followed by Si, Ti, and C. Because SiO_2_, PDMS, and EP in the coating contain a large number of oxygen elements, and laser etching leads to the oxidation of the titanium alloy surface, the distribution of the O element is the most. The Si element mainly comes from PDMS and SiO_2_, which are evenly distributed on the surface. At the same time, the Ti element is only distributed in the exposed part, because the EDS element analysis detects shallow surface elements by exciting element characteristic X-ray. The C element is mainly provided by PDMS and EP, and is also evenly distributed on the surface. As shown in Figure 4b, the energy spectrum results of elemental analysis on the sample surface are shown. Therefore, PDMS and EP play an essential role in the coating, adhering to SiO_2_ particles and reducing the surface energy, to improve the superhydrophobic performance of the surface with the micro–nano composite structure.

### 3.2. Surface Wettability Test

The static CA is an important parameter to characterize the wettability. Generally, the superhydrophobic surface refers to the surface with a WCA greater than or equal to 150°. The surface of unmodified titanium alloy belongs to a hydrophobic surface, and WCA can reach about 118°on the polished smooth surface, while the surface of the sample after laser etching treatment quickly changes to a superhydrophilic surface, which is often due to the oxidation of the titanium alloy surface caused by laser etching, which increases the surface energy. To make the surface of titanium alloy have good superhydrophobic properties, low surface energy coatings were deposited on the surface, and nanoparticles were added to increase the surface roughness and better support the droplets to form a superhydrophobic surface.

By testing the WCA and SA of different samples, the data presented in Figure 5 were obtained. During the experiments, the ambient temperature was approximately 26 °C, and the water droplet volume was around 2 μL. It can be clearly observed that the WCA of all coated samples exceeds 150°, meeting the criterion for superhydrophobicity. However, only Samples 2, 3, 5, 6, 8, 9, and 12 exhibit an SA below 10°. A comprehensive evaluation of surface wettability based on both WCA and SA confirms that only these seven samples can be classified as superhydrophobic surfaces. Separately, the static CAs of all samples are close to 153°, and the difference between them is slight. The results show that the prepared coating has good water repellency, and PDMS, EP, and SiO_2_ play an essential role in the coating. The combination of PDMS and EP provides low surface energy and adhesion properties for the coating. SiO_2_ particles act as micro/nanostructures on the surface of the titanium alloy, which makes the droplets maintain the Cassie–Baxter state on the surface.

It can be observed that the SA values of Samples 1, 5, 7, 10, and 11 are relatively high. A further analysis of the additive content in the coating solution reveals that surfaces with a higher concentration of SiO_2_ particles facilitate easier droplet rolling. This indicates that increasing the SiO_2_ content significantly reduces surface adhesion, thereby lowering the SA. Therefore, it can be concluded that the amount of SiO_2_ nanoparticles plays a crucial role in determining the SA and overall dynamic wettability of the coated surface. A higher concentration of SiO_2_ particles enables a more effective filling of surface features such as pits, resulting in a more complex micro–nano composite structure. This enhanced morphology better supports droplets and promotes the formation of a gas–liquid composite interface, which reduces the actual contact area and adhesion between the droplets and the surface. Consequently, droplets can slide more easily during surface inclination.

The above WCA and SA data demonstrate that the composite coating consisting of PDMS, EP, and SiO_2_ exhibits excellent superhydrophobic properties. A correlation analysis of the effects of PDMS, EP, and SiO_2_ contents on WCA and SA reveals that SiO_2_ particles have the most significant influence. An increase in SiO_2_ content slightly enhances the WCA but has a more pronounced effect on reducing the SA. EP also affects the WCA to some extent, with higher EP content causing a slight decrease in WCA. In contrast, the PDMS content has minimal impact on both WCA and SA values. It can be inferred that the hydrophobic structure formed by the adhesion of PDMS and SiO_2_ particles has good stability, and the increase in PDMS content has little effect on the wettability. The increase in EP content may cause the surface of SiO_2_ particles to be covered by EP, affecting the hydrophobic effect of PDMS with low surface energy on the surface. Finally, the content of SiO_2_ particles affects the complex nanostructures formed on the surface. According to the Wenzel model, it has a significant influence on the wettability of the surface.

### 3.3. Mechanical Stability Test of Coating

Due to the particularity of the surface, the abrasion of superhydrophobic coatings often leads to a decline or even loss of hydrophobic properties. When evaluating the overall performance of the coating, mechanical stability is as important as the superhydrophobic function. To investigate the influence of abrasion on the surface hydrophobic coating, the WCA was measured after the sample underwent abrasion on sandpaper over different distances under the weight of the sample, as shown in Figure 6a. The applied weight during the test was 200 g, and the abrasive material was 1200 cw silicon carbide (SiC) sandpaper. The sample was dragged slowly at an average speed during the test, with a total abrasion distance of 1000 cm.

Figure 6b shows the surface morphology of the sample before and after abrasion. It can be observed that there are many SiO_2_ particles on the surface before the abrasion test, as well as protrusions of the melted and re-solidified titanium alloy. After abrasion, part of the SiO_2_ particles covering the surface were worn away, but there were still many SiO_2_ particles on the surface, especially a large number of particles in the pits. This is due to the storage capacity of the pits, which keeps particles from falling off during abrasion. In the part circled by the ellipse in the figure, there are flakes of SiC particles on the abrasive paper. This indicates that the SiC particles on the abrasive paper are accumulated and embedded in the sample pit during abrasion. Noticeable abrasion marks can be observed on the surface of the sample. The enlarged part of Figure 6b shows the marks of the molten bulge worn by the abrasive paper, and there are evident scratches on the surface. The base bulge of the melted and solidified titanium alloy plays a supporting role in the abrasion process, forming a solid armor structure, which makes the abrasion contact first, improves the abrasion resistance, and reduces the loss of coating particles. The primary damage mechanism caused by friction on the coating is cutting or spalling abrasion. The localized detachment of SiO_2_ particles progressively undermines the multi-scale hierarchical structure, directly resulting in a reduction of the WCA. As shown in Figure 6c, the WCA of the sample decreases noticeably after abrasion, with the surface generally retaining hydrophobicity. It is evident that samples with a higher concentration of SiO_2_ particles tend to exhibit a more significant decline in WCA following abrasion.

Table 2 presents the variation in WCA before and after sample abrasion. Based on a comparative analysis of the quantities of EP and PDMS, it can be inferred that both components exert specific effects on the surface wettability of the samples. An analysis of the WCA reduction indicates that when the EP content is 2 g, the decrease in surface WCA is generally smaller, suggesting enhanced abrasion resistance. In contrast, variations in PDMS content show minimal influence on the WCA of the abraded surfaces. This may be because when the amount of PDMS is 1 g, the SiO_2_ particles on the surface have been completely wrapped, and the hydrophobic effect has reached the optimal value. Even increasing the amount of PDMS will not have a greater impact on the hydrophobic effect. However, EP plays a critical role in influencing the bonding strength of SiO_2_ particles. When the EP content is limited to 1 g, it is challenging to achieve strong adhesion between the particles and the substrate surface. In contrast, an EP dosage of 2 g significantly enhances the bonding strength both among the particles and between the particles and the substrate, thereby reducing particle detachment during abrasion and contributing to an improved retention of surface hydrophobicity. Overall, Sample 11 exhibits superior mechanical stability with respect to its WCA.

Furthermore, the mechanical stability of the coatings was assessed through sandpaper abrasion tests. Although the water contact angle (WCA) decreased after 1000 cm of abrasion, coatings with a high concentration of SiO_2_ particles retained strong superhydrophobicity. This observation aligns with the findings of Shen et al. [30], who fabricated superhydrophobic coatings via composite electrodeposition. Their results showed that, despite a reduction in WCA from 159° to 146° after abrasion, the coating maintained a relatively high contact angle, indicating good abrasion resistance. Similarly, He et al. reported excellent mechanical stability in superhydrophobic coatings prepared by a simple spray-coating method. In particular, a coating with 6.7 wt% SiO_2_ exhibited a WCA of 159.6°, which decreased to 146.5° after sandpaper abrasion, demonstrating outstanding wear resistance and mechanical durability [31]. These results are in close agreement with the findings of the present study, further confirming that higher SiO_2_ concentrations enhance the mechanical robustness of superhydrophobic coatings.

### 3.4. UV Durability Test of Coating

The purpose of the UV durability test is to evaluate the stability and durability of a superhydrophobic metal substrate surface coated with an SiO_2_ composite coating in a UV environment. In practical use, superhydrophobic surfaces face the coupling effect of multiple factors such as light radiation, water vapor, oxygen, and temperature changes in the natural environment. Ultraviolet radiation may especially lead to the photooxidative degradation of organic-modified groups. Therefore, the performance evolution process of the coating in the natural environment is accelerated by the ultraviolet aging test. The test is conducted by continuous irradiation with a UV lamp at room temperature, and the WCA of the surface is measured at different times of irradiation.

The test results are shown in Figure 7a. The WCA of both sample surfaces decreased significantly after 24 h of UV irradiation. Notably, the decline in WCA for Sample 3 was less pronounced than that of Sample 1. With increasing irradiation time, the WCA of Sample 3 gradually decreased but remained around 150° after 24 h, maintaining its superhydrophobic properties. In contrast, the WCA of Sample 1 decreased rapidly over time, ultimately dropping to approximately 132°, indicating a substantial loss in hydrophobic performance. This result shows that the surface superhydrophobic property with high SiO_2_ particle content has stronger resistance to ultraviolet light. It is speculated that the more SiO_2_ particles, the greater the thickness of the surface coverage. This can prevent ultraviolet rays from penetrating deeper layers in the process of radiation, and still provide good physical support for the attachment of deep SiO_2_ particles and the existence of organic hydrophobic groups. On the other hand, in the high-intensity ultraviolet environment, the organic hydrophobic groups are bound to undergo a certain degree of photo-oxidative decomposition and other reactions. Still, the multiple complex structures formed by SiO_2_ particles and surface pits make the coating able to resist the degradation of hydrophobic properties caused by aging.

Microscopic observation and elemental distribution analysis were performed on the sample surface after UV irradiation. As shown in Figure 7b, the elemental distribution of Sample 1 is presented. It can be found that UV irradiation will not affect the microstructure, but the element distribution on the surface has changed dramatically. Compared with the non-irradiated surface, the content of O increased significantly, while the content of Si decreased significantly. The content of other elements, such as Ti and C, did not change significantly. The change in surface element distribution showed that some organic matter on the surface had been oxidized and decomposed after UV irradiation. Therefore, UV radiation has a certain impact on the surface coating, but the surface still has good hydrophobic properties. In addition, the increase in SiO_2_ particles can reduce the effect of ultraviolet radiation on the decrease in WCA, thereby maintaining the surface’s superhydrophobic state. These results are consistent with the studies by Wang et al. [32] and Zong et al. [33], both of which indicated that coatings with a higher SiO_2_ content exhibit better UV stability and photocatalytic performance under UV exposure. Specifically, Wang et al. found that coatings with a higher content of SiO_2_ particles are more effective in maintaining superhydrophobicity and remain stable for a longer period. Similarly, this study also found that coatings with a higher SiO_2_ content exhibit stronger resistance to degradation under UV radiation, thereby effectively maintaining their hydrophobicity. In conclusion, the prepared superhydrophobic composite coating on a titanium alloy substrate has certain UV durability, and the surface UV resistance can be further enhanced by increasing the SiO_2_ particle content.

### 3.5. Surface Adhesion and Self-Cleaning Test

Although the WCA and SA can reveal the water repellent ability of the surface from a macro perspective, the adhesion force between the droplet and the solid substrate can more accurately characterize the performance of the surface in practical applications. The self-cleaning effect directly related to the service quality and durability of the surface in a dusty and polluted environment. In this study, the adhesion and self-cleaning effect of the surface to droplets was tested, and the comprehensive properties of the surface were further investigated.

The surface adhesion is characterized by the degree of adhesion of the droplets on the surface. The droplets are extruded through the needle tube, and the droplets remain on the needle through intermolecular cohesion and electrostatic adsorption. Then, the needle tube is lowered to make the droplets contact the surface, and finally, the needle tube is lifted to observe the adhesion effect of the droplets on the surface in this process. As shown in Figure 8a, when the droplet contacts the surface, it remains highly spherical. After the droplet is wholly separated from the surface, no droplet residue is observed, so there is almost no adhesion on the prepared coating surface. The self-cleaning effect is usually used to simulate the removal effect of rainwater or water flow on pollutants in the natural environment, and it is an essential means to test the surface functionality. During the test, the sample is tilted at a certain angle, carbon black powder is dispersed on the surface of the sample, and then the drip tube is used to drip water on the surface to make the water droplets roll or flow on the surface. The carrying and removal degree of carbon black powder by water is observed. As shown in Figure 8b, the carbon black powder on the surface of the sample is almost entirely carried away by the droplets, which indicates that the sample surface has good self-cleaning characteristics. Therefore, the prepared superhydrophobic coating on the titanium alloy substrate can not only maintain high WCA and low adhesion, but also enable water droplets to carry and quickly remove the attached pollution particles when rolling on the surface, showing excellent self-cleaning and decontamination ability. This ability can not only effectively reduce the dependence on manual cleaning and chemical reagents, but also significantly improve the service life and maintenance efficiency of equipment surfaces in harsh application environments such as aviation, aerospace, and marine engineering, which is of great significance to achieve emission reduction and consumption reduction and improve the reliability of use.

## 4. Conclusions

In this study, an array of pits was formed on the titanium alloy substrate by a laser, and a kind of superhydrophobic material with strong performance was prepared on the EP@PDMS@F-SiO_2_ composite coating surface. The influence of different proportions of composite coating solution on the surface wettability was investigated. The main conclusions are as follows: (1) The microstructure observation shows that SiO_2_ particles in the coating can be evenly distributed on the pit surface. Moreover, EP and PDMS, with adhesion and hydrophobicity properties, are wrapped onto the surface of SiO_2_ particles, which makes it easier for SiO_2_ particles to adhere to the titanium alloy substrate, forming a micro–nano composite hydrophobic surface structure. (2) The prepared coating surface has good superhydrophobic properties; EP improves the adhesion of particles and PDMS reduces the surface energy. In contrast, SiO_2_ particles fill pits and gaps to form a complex structure to support the surface droplets. (3) Through analysis, the optimal composition of the superhydrophobic composite coating on the titanium alloy surface was determined to be EP 2 g/PDMS 2 g/F-SiO_2_ 1 g (Sample 11). This formulation exhibits outstanding performance, with an initial WCA of 153.08° and the smallest reduction in WCA after undergoing a 200 g load abrasion test over 1000 cm, showing only a slight decrease of 5° to 148.08°. (4) This solid titanium alloy substrate coating structure shows good mechanical stability and UV durability. (5) The low adhesion of the coating surface to droplets gives the surface excellent self-cleaning ability for maintaining the clean state of the surface. In conclusion, the superhydrophobic composite coating on the titanium alloy substrate prepared in this study maintains good mechanical stability, UV durability, and hydrophobicity, indicating that the surface exhibits high potential for practical applications, and lays a more solid research foundation for the design and practical promotion of a functional metal substrate.

## Figures and Tables

**Figure 1 biomimetics-10-00404-f001:**
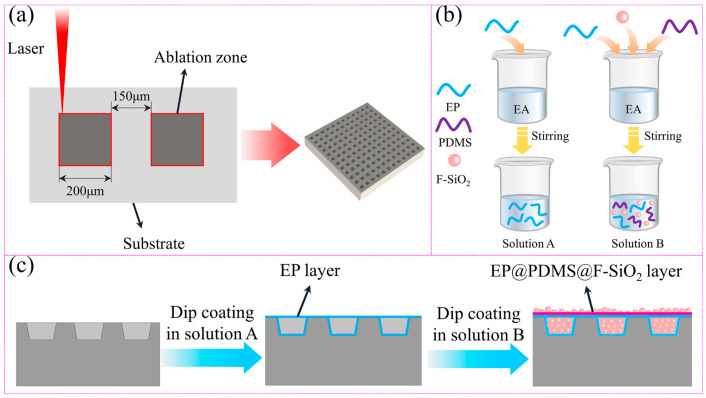
Schematic diagram of preparation process of composite coating on superhydrophobic titanium alloy surface: (**a**) laser etching surface size parameters, (**b**) preparation process of solutions A and B, and (**c**) schematic diagram of the preparation of composite coatings.

**Figure 2 biomimetics-10-00404-f002:**
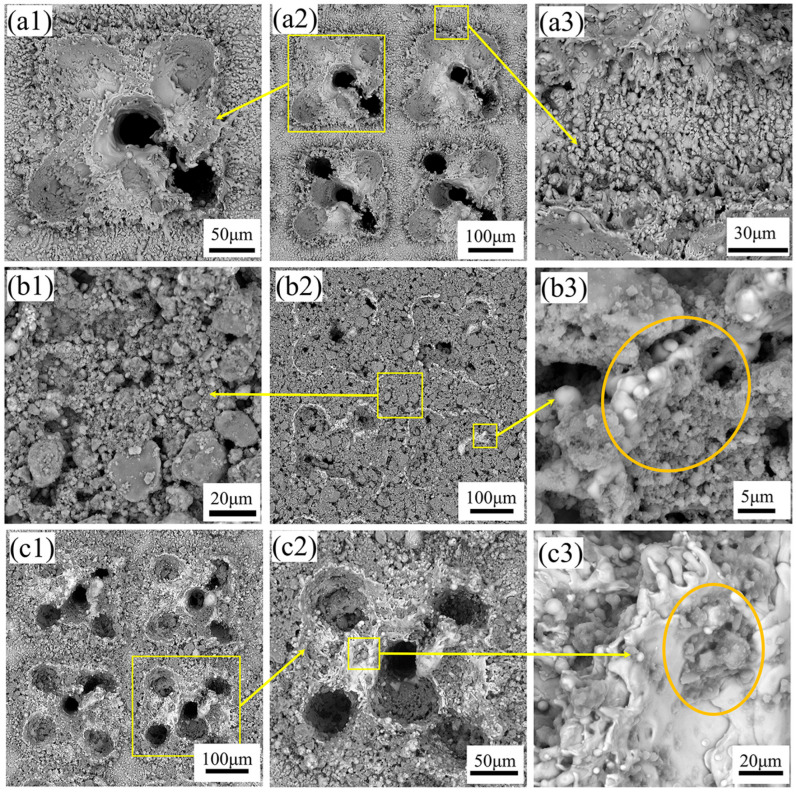
SEM images of sample surfaces: (**a1**,**a3**) are magnified views of two different regions in (**a2**), showing the laser-etched surface microstructure, (**b1**,**b3**) are magnified views of two different regions in (**b2**), showing the surface morphology of Sample 12, and (**c1**–**c3**) are progressively magnified local areas of Sample 1, showing surface particle distribution.

**Figure 3 biomimetics-10-00404-f003:**
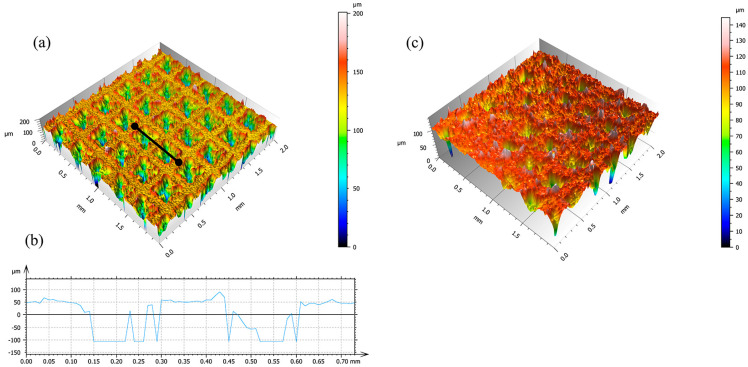
Three-dimensional surface morphology of the samples: (**a**) laser-etched pit array, (**b**) cross-sectional profile curve, and (**c**) coated surface of Sample 3.

**Figure 4 biomimetics-10-00404-f004:**
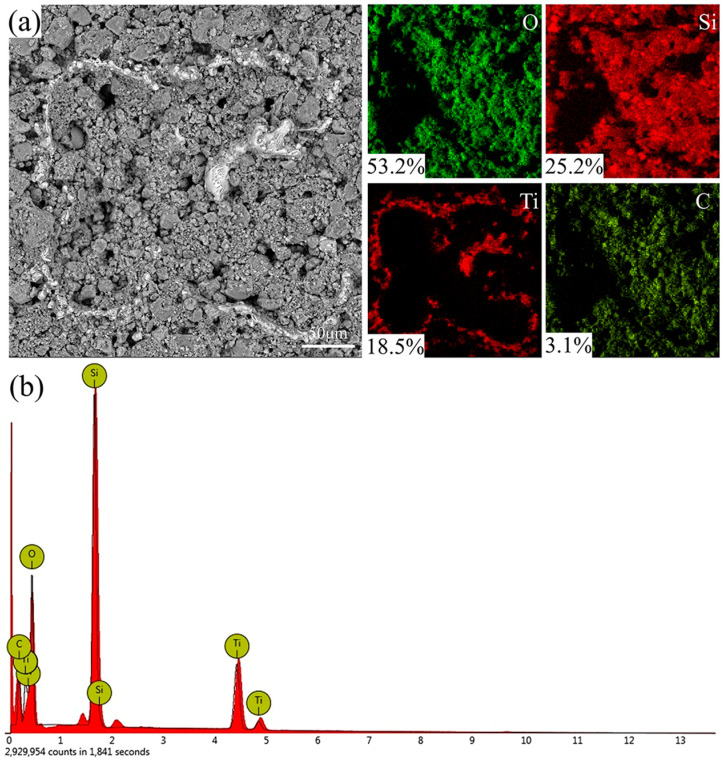
Surface element analysis of sample: (**a**) surface element distribution and (**b**) EDS spectrum.

**Figure 5 biomimetics-10-00404-f005:**
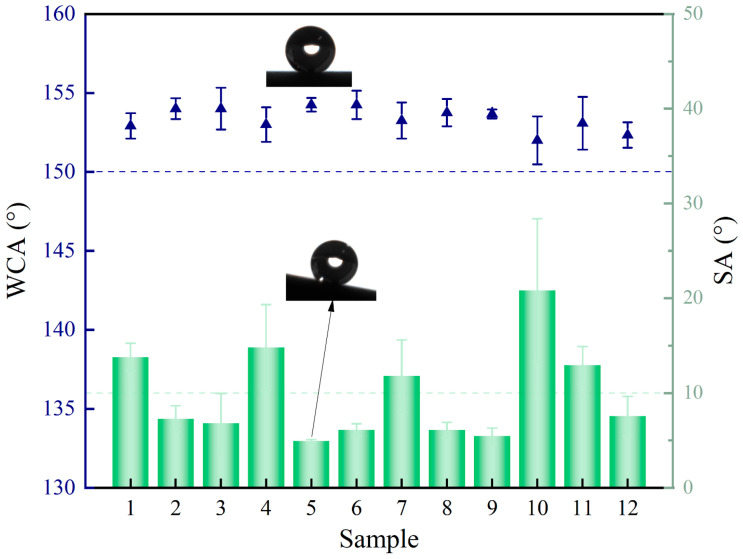
WCA and SA of a sample surface.

**Figure 6 biomimetics-10-00404-f006:**
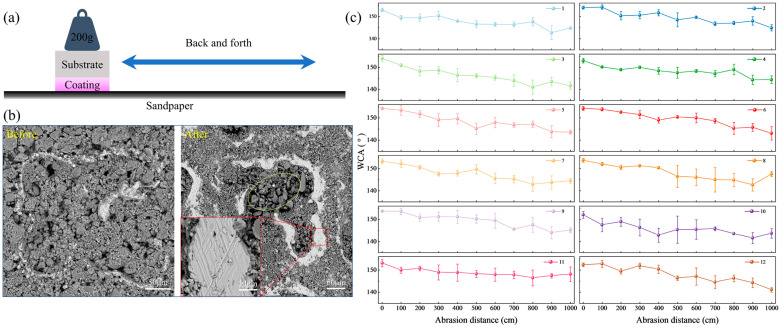
Test schematic and results: (**a**) abrasion test using sandpaper, (**b**) surface morphology of Sample 6 before and after abrasion, and (**c**) variation in WCA with abrasion distance.

**Figure 7 biomimetics-10-00404-f007:**
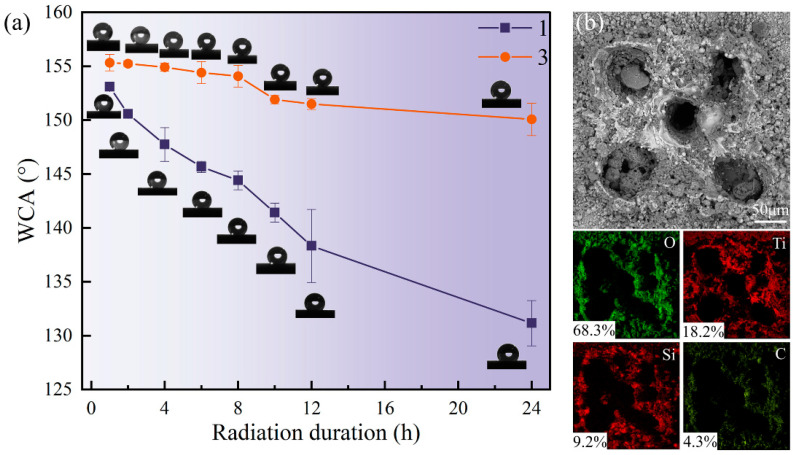
Schematic diagram of WCA variation with irradiation time and elemental distribution: (**a**) variation in WCA of Samples 1 and 3 with irradiation time and (**b**) surface elemental distribution of Sample 1 after ultraviolet irradiation.

**Figure 8 biomimetics-10-00404-f008:**
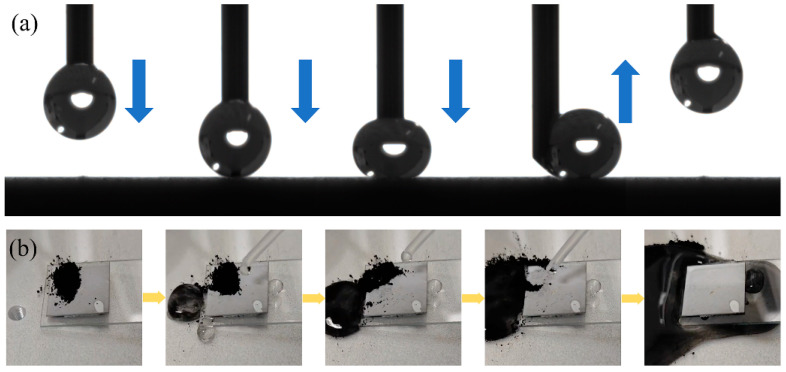
Schematic diagram: (**a**) surface low adhesion effect and (**b**) self-cleaning characteristics.

**Table 1 biomimetics-10-00404-t001:** Preparation of superhydrophobic coatings on titanium alloy substrates.

No.	EP/g	PDMS/g	SiO_2_/g
1	1	1	0.5
2	1	1	1
3	1	1	1.5
4	1	2	0.75
5	1	2	1.25
6	1	2	2
7	2	1	0.75
8	2	1	1.25
9	2	1	2
10	2	2	0.5
11	2	2	1
12	2	2	1.5

**Table 2 biomimetics-10-00404-t002:** Variation in average water contact angle of different samples after 1000 cm of surface abrasion.

No.	Initial WCA/°	WCA After Abrasion/°	Reduction of WCA/°
1	152.92	144.83	8.09
2	154	144.92	9.08
3	154	141.67	12.33
4	153	144.12	8.88
5	154.25	143.58	10.67
6	154.25	143.08	11.17
7	153.25	144.5	8.75
8	153.75	147.5	6.25
9	153.67	145.08	8.59
10	152	143.67	8.33
11	153.08	148.08	5
12	152.33	141.08	11.25

## Data Availability

Data will be made available on request.

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
