# Peer review of "Fabrication of EP@PDMS@F-SiO2 Superhydrophobic Composite Coating on Titanium Alloy Substrate"

_biomimetics, 2025, doi:10.3390/biomimetics10060404_

Round 1
Reviewer 1 Report
Comments and Suggestions for Authors
This study develops a superhydrophobic EP@PDMS@F-SiOâ‚‚ composite coating on titanium alloy (Ti-6Al-4V) using laser-etched micro-pit arrays. Optimizing ratios of epoxy resin (EP), polydimethylsiloxane (PDMS), and fluorinated nanosilica (F-SiOâ‚‚) via orthogonal experiments enhanced wettability, mechanical stability, and UV durability. The laser-created micro-nano structure securely anchored particles, achieving water-repellency (contact angle >150°). The coating resisted abrasion (200g load, 1000cm wear) and UV degradation (higher F-SiOâ‚‚ improved durability). It also exhibited low adhesion, self-cleaning, and contaminant removal. This eco-friendly, non-fluorinated approach offers practical insights for durable superhydrophobic surfaces in engineering applications. But some points should be revised:
- sample 1 and 11 are the same. Authors should be consider the right percentage of materials.
- Use abbreviations such as SEM instead of Scanning electron microscopy. please check for other words.
- Figure 2 is a bit confusing. Also, the scale bar of the images are not the same, and the caption of the figure is not appropriate .
- The authors had different samples to examine, so it's necessary to show the process of microstructural changes for different samples. While only two samples were examined.
- Sample 1 and 11 are similar, so it's expected that their properties will not differ.
- In Figure 6. "(b) Distribution of surface elements after ultraviolet irradiation." which sample does it belong to?
- In the results and discussion section, only the results are reported and no proper discussion is made, nor is any comparison provided with other valid references. The authors should make a basic revision in this section.
Reviewer 2 Report
Comments and Suggestions for Authors
Manuscript is of interes, but requires significant revisions.
Is it notation EP@PDMS@F-SiOâ‚‚? I mean, is symbol "@" really necessary? May be i just do not know something, but it is a bit confusing to see "@" in the title and within the text. May be it is some sort of computer mistake and there should be something like "-"?
Line 99. FAS abbreviation should be explained.
Line 108-109. "The mixed solution in this experiment is divided into a and B." I guess, "a" should be capital.
Figure 1 caption does not contain description, what are (a), (b), etc.
Wettability tests are not mentioned in "Materials and Methods" paragraph.
Paragraph 3.1. Authors write, that after coating treatment the surface is covered with nanoparticles and after it is written that those particles are SiO2. But the size of used initial SiO2 particles was not mentioned in "Materials and methods" part. So it is hard to understand, wether those particles were subjected to some transformations during coating treatment.
Figure 4. SA abbreviation should be explained, while in fig. caption it is written "rolling angle of a sample surface". May be it is RA instead of SA?
Table 2 shows very accurate values and there are no any error data. There is no information about statistical significance of represented data. This is recommended to be commented and corrected.
There is no discussion at all. Authors should discuss the obtained results, compare them with the data obtained by other authors, explain, why all the mentioned processes occur.
Comments on the Quality of English LanguageThere are some misprints.
Reviewer 3 Report
Comments and Suggestions for Authors
1) Lines 48-53: Please add more precise characterization values to compare the listed superhydrophobic design methods with literature references to support this statement.
2) Literature review should also be expanded to confirm the outstanding properties of SiO2 nanoparticles and its modifications. Are there analogs of nanoparticles or other materials? State the advantages of SiO2 nanoparticles within the framework of comparative analysis of material properties.
3) Please explain why square grooves were chosen and with such a matrix pitch. Is the depth of the grooves known? Have you conducted research on the influence of the matrix on the mechanical properties of the sheets?
4) Line 120: Write in detail what kind of sandpaper was used to prepare the surface? Did you carry out the polishing process to a mirror surface? What does “smooth” mean?
5) Table 1. Preparation of Superhydrophobic Titanium Alloy Specimens: Requires rewordings such as: “Preparation of superhydrophobic coatings on titanium alloy substrate”.
6) It is difficult to distinguish the scale bar in the SEM images. Please color it differently
7) Figure 3b: low quality image
8) The conclusion does not indicate which of the solution ratios is optimal.
CONCLUSION: After minor revisions have been made, the article may be accepted for publication.
Round 2
Reviewer 1 Report
Comments and Suggestions for Authors
The revise version is acceptable.